# Co-Existence of Free-Living Amoebae and Potential Human Pathogenic Bacteria Isolated from Rural Household Water Storage Containers

**DOI:** 10.3390/biology10121228

**Published:** 2021-11-25

**Authors:** Natasha Potgieter, Clarissa van der Loo, Tobias George Barnard

**Affiliations:** 1One Health Research Group, Department of Biochemistry and Microbiology, Faculty of Science, Engineering and Agriculture, University of Venda, Thohoyandou 0950, Limpopo Province, South Africa; 2Water and Health Research Centre, Doornfontein Campus, University of Johannesburg, Johannesburg 2112, Gauteng, South Africa; tgbarnard@uj.ac.za (T.G.B.); clarissakruger@gmail.com (C.v.d.L.)

**Keywords:** biofilm, borehole water, free-living amoebae, household water storage containers, potential human pathogenic bacteria, rural environment

## Abstract

**Simple Summary:**

In many households in rural communities, water needed for drinking and cooking is fetched from rivers, fountains, or boreholes shared by the community. The water is then stored in various storage containers for several days without treatment and exposed to several conditions that could potentially contaminate the water and cause diseases. If the storage containers are not regularly and properly cleaned, biofilms can form inside the containers. Several microorganisms can be found inside the biofilm that can potentially cause diseases in humans. One such group of organisms is called free-living amoebae, which graze on the bacteria found inside the biofilm. Several of these potentially harmful bacteria have adapted and can survive inside these free-living amoebae and potentially cause diseases when ingested by humans.

**Abstract:**

This study investigated the co-existence of potential human pathogenic bacteria and free-living amoebae in samples collected from stored water in rural households in South Africa using borehole water as a primary water source. Over a period of 5 months, a total of 398 stored water and 392 biofilm samples were collected and assessed. Free-living amoebae were identified microscopically in 92.0% of the water samples and 89.8% of the biofilm samples. A further molecular identification using 18S rRNA sequencing identified *Vermamoeba vermiformis*, *Entamoeba* spp., *Stenamoeba* spp., *Flamella* spp., and *Acanthamoeba* spp. including *Acanthamoeba* genotype T4, which is known to be potentially harmful to humans. Targeted potential pathogenic bacteria were isolated from the water samples using standard culture methods and identified using 16S rRNA sequencing. *Mycobacterium* spp., *Pseudomonas* spp., *Enterobacter* spp., and other emerging opportunistic pathogens such as *Stenotrophomonas maltophilia* were identified. The results showed the importance of further studies to assess the health risk of free-living amoebae and potential human pathogenic bacteria to people living in rural communities who have no other option than to store water in their homes due to water shortages.

## 1. Introduction

Statistics South Africa [1] reported that only 88.2% of households have access to an improved drinking water source in South Africa. Many households in rural and peri-urban areas of South Africa have no other choice but to store water to use for drinking, cooking, and other household chores [2]. Water quality deteriorates during household storage due to physical, chemical, and microbiological processes [2,3,4,5,6]. If the storage containers are not properly or regularly cleaned and hygiene behaviours are poor inside the household, a biofilm forms inside the container [7,8,9].

The presence of biofilms in water storage containers is a major contribution in the survival and transmission of free-living amoebae (FLA) and potential amoeba-resistant bacteria (ARB)—which include human pathogens—in stored water containers, especially in large water storage tanks [10].

Biofilms can harbour potentially pathogenic microorganisms that can cause infections if not removed properly [11]. Amoebae are found in biofilms due to their ability to strongly adhere to the sub-stratum through their large area of contact [12] and they are also able to graze on the bacterial biofilm [13]. The deterioration of stored water [14,15] and the formation of biofilms within water storage containers [9,16,17] have been investigated; many of these studies have been in low-income countries [18,19,20,21] but very few studies have been done in South Africa. In 2002, Momba and Kaleni [22] demonstrated the direct relationship between the material of the storage container, the quality of the intake water, and the degree of bacterial regrowth. They found that total coliforms grew at a higher rate on polyethylene than on galvanised steel; however, steel is vastly more expensive than plastic and most people living in rural communities can only afford plastic storage vessels [22], of which most are old used food or paint containers. Other studies in South Africa have investigated the composition, biomass, adhesion mechanisms, and resistance profiles of biofilms in water storage containers but the focus has been on total coliforms, *E. coli, Salmonella, Clostridium perfringens* [22], somatic coliphages [3], heterotrophic bacteria [17,23] and faecal coliforms [23,24].

Unfortunately, there is very little literature on studies carried out on FLA biofilms in drinking water storage containers globally. Taravaud et al. [25] found a marked increase in the FLA biofilm density at the top levels of drinking water storage towers. Stockman et al. [26] enumerated FLA in (municipal) household water. Rohr et al. [27] and Muchesa et al. [28,29] focused on FLA in hospital water networks; however, most researchers focus on the bacterium *Legionella pneumophila* and its interaction with FLA within biofilms [30,31,32,33,34] rather than on other known ARBs or human pathogens. The aim of this study was to investigate the prevalence of FLA and potential human pathogenic bacteria in stored household water as well as the prevalence of FLA present in biofilms within household water storage containers used in South African rural households.

## 2. Materials and Methods

### 2.1. Study Area

A total of 82 households in seven informal settlements on the border between the Gauteng and Mpumalanga provinces in South Africa were randomly selected and included in this study. All households used borehole water as their primary water source, collected from either a central free-standing tap outside the home, a windmill, or a large water storage tank (with a capacity of between 260 L and 10,000 L). Distances to the water collection points from the households varied between the settlements and ranged from a few meters to approximately one kilometre. Households would typically collect drinking water using 20 L plastic buckets that were reused food-grade plastic buckets that previously contained cooking oil. The buckets were cleaned and repurposed by the households to store water for drinking, cooking, and washing purposes (Figure 1A,B).

### 2.2. Sample Collection

All samples were collected once a month from each study household over a period of 5 months between March and July of 2016. Water was collected from the 20 L household storage buckets using the cup/jug that the household normally used for water retrieval (Figure 1A) and transferred into sterile 1000 mL collection containers that were clearly marked with the settlement name, house number, and date. Biofilm samples were collected by swabbing an area of 10 × 10 cm on the inside of the water storage container with a sterile swab (COPAN, Murrieta, CA, USA). The swab was inserted into the water under the waterline, as far down as possible (approximately 15 cm) without the person sampling touching the water with their hand (Figure 2). The swab was placed back in its container and clearly marked with the settlement, house number, and the date collected. All samples were refrigerated at 4 °C and transported to the laboratory early the next morning for the analysis.

### 2.3. Isolation and Identification of Potential Human Pathogenic Bacteria

Amoebal enrichment suspensions (100 µL of each sample) were inoculated onto different selective media for the isolation of potential harmful bacteria The study only focused on potential harmful bacteria including *Salmonella, Shigella, Legionella, Mycobacterium, E. coli, Vibrio*, and Coliform bacteria based on a review article of ARB by Greub and Raoult [35]. The selective media types used to isolate these bacteria included a BrillianceTM *E. coli*/coliform selective medium for *Escherichia coli* and other coliform bacteria (Oxoid CM1046B); a xylose lysine deoxycholate (XLD) agar for *Salmonella* and *Shigella* spp. (Oxoid CM0469B); a thiosulphate citrate bile salts sucrose (TCBS) agar for pathogenic *Vibrio* spp. (Oxoid CM0333B); a charcoal yeast extract (CYE) agar (Oxoid CM0655B) with a *Legionella* buffered charcoal yeast extract (BCYE) growth supplement for *Legionellae* spp. (Oxoid SR0110A); and Middlebrook 7H9 media with an oleic albumin dextrose catalase (OADC) growth supplement for *Mycobacterium* spp. For each type of selective media, positive controls were used to ensure that the media would serve as an accurate source of information regarding the unknown organisms cultured on them and included *Vibrio cholerae* O139 (NSCC), *Salmonella enteritidis* (NSCC), *Shigella boydii* (NCTC 9329), *Escherichia coli* (ATCC 43888), *Legionella pneumophila* (ATCC 33152), and *Mycobacterium avium* (NSCC). All incubations were done as per the specifications of the manufacturers. The Ziehl–Neelsen stain [36] was used to confirm *Mycobacterium* spp. Non-specific colonies on the selective media were picked and re-plated onto nutrient agar (Sigma-Aldrich 70148) to isolate single colonies for further identification. Likewise, single colonies identified using selective media as presumptive *Mycobacterium, Shigella, Legionella, Salmonella, Shigella, E. coli*, or coliform species were also picked and streaked on nutrient agar for purification before a molecular identification. All molecular identifications were performed by Inqaba Biotechnical Industries (Pty) Ltd. (Pretoria, South Africa). Only the 16S rRNA gene was investigated using primers sourced from the literature. These primers amplify almost the entire length of the gene [37] and the target region is conserved in many bacterial species [38]. For the *Mycobacterium* spp. identification, the forward primer pA (5′-AGAGTTTGATCCTGGCTCAG-3′) and the reverse primer MSHE (5′-GCGACAAACCACCTACGAG-3′) were used [39]; for *Legionella* spp., the forward primer LEG 225 (5′-AAGATTAGCCTGCGTCCGAT-3′) and the reverse primer LEG 858 (5′-GTCAACTTATCGCGTTTGCT-3′) were used [40]; and for the identification of the colonies from the nutrient agar, XLD, and Brilliance *E. coli*/coliform media, 16S rRNA primer pairs that included the forward primer 27 (5′-AGAGTTTGATCMTGGCTCAG-3′) and the reverse primer 1492 (5′-CGGTTACCTTGTTACGACTT-3′) were used [41].

### 2.4. Isolation of FLA from the Water Samples

The FLA were then isolated using the amoebal enrichment method described by Thomas et al. [42]. All water samples (500 mL of each sample) were filtered through 0.45 µm nitrocellulose filters (Millipore, Burlington, MA, USA) with a filter manifold (Sartorius, Goettingen, Germany). The filter was aseptically placed face-side down onto a non-nutrient agar (NNA) plate and covered with a layer of heat-killed *Escherichia coli* (type strain, ATCC 25,922), which served as a food source for the amoeba; a few drops of Page’s amoebal saline (PAS) were then added to aid mobility and the plates were incubated aerobically at 33 °C [42]. The NNA plates were checked daily for the appearance of amoebal cysts using a 10× light microscope. A disposable Pasteur pipette was used to cut small plugs from the plates; these plugs were placed onto new NNA plates (covered with heat-killed *E. coli*) and one drop of PAS was added to each plug. This process purified the amoebae as other organisms remained on the original plate. These were sub-cultured onto new NNA plates with *E. coli* and PAS until the culture was axenic. The sub-cultured plates were flooded with 2 mL of PAS and a sterile plastic loop was used to gently scrape the axenic amoebae from the plate surface. This suspension was transferred to clearly marked sterile 2 mL Eppendorf^®^ tubes. Before freezing the 2 mL tubes for a later analysis, 1 mL of the amoebal suspension was lysed by passaging the suspension through a 27-gauge syringe and vortexed at 2500 rpm (Vortex Genie^®^ 2-Mixer 240 V, 50 Hz Scientific Industries, Bohemia, New York, NY, USA) to release the ARB. The lysed amoebae were stored in a different sterile marked tube at −70 °C.

### 2.5. Isolation of FLA from the Biofilm Samples

Each biofilm swab was vortexed in its individual sterile tube for 30 s with 1 mL of Page’s amoebal saline (PAS) and then centrifuged for 10 min at 800× *g*. Of the supernatant, 200 µL was inoculated onto a 24-well plate and the pellet was suspended in 100 µL PAS and plated onto a non-nutrient agar (NNA) plate seeded with heat-killed *E. coli*. The 24-well plate was centrifuged for 30 min at 1500× *g* and incubated aerobically at 33 °C for three days as per Thomas et al. [42] with the following exceptions: after the three days, when the amoebae had attached to the plate, the supernatant was carefully removed (without disturbing the cells) and discarded. A fresh volume of 100 µL PAS was added to the layer of FLA cells in the 24-well plates and passaged by pipette until well-mixed then again incubated aerobically at 33 °C for three days. The 24-well plate co-cultures were washed, as previously described, and read by an inverted microscope to observe the FLA in their natural state.

### 2.6. FLA Identification

Only a subset (±10%) of the isolates that were positive for FLA by inverted microscopy were randomly selected from the samples containing FLA and *Acanthamoeba* cysts to send for an 18S rRNA PCR and sequencing, which were performed by the Department of Hygiene, Social, and Environmental Medicine at Ruhr University, Bochum, Germany.

The selected samples stored after the initial amoebal enrichment were freshly inoculated onto NNA plates and allowed to grow. The plates were sealed individually with Parafilm M^®^ before they were packaged and sent to our collaborators in Germany for PCR and sequencing. Amoebal DNA was extracted from 200 μL of the prepared amoebae suspension in the amoebae-positive plates using a QIAamp DNA Blood Mini Kit (Qi-agen, Hilden, Germany) as per manufacturer’s protocol. The nucleic acid was eluted in 100 μL of an elution buffer into a sterile clearly marked 1.5 mL microcentrifuge tube and stored at −20 °C. The FLA genotyping was performed using the forward primers Ami6F1 (5′-CCAGCTCCAATAGCGTATATT-3′) and Ami6F2 (5′-CCAGCTCCAAGAGTGTATATT-3′) as well as reverse primer Ami9R (5′-GTTGAGTCGAATTAAGCCGC-3′) to amplify the 18S rRNA gene [42]. For the *Acanthamoeba* genotype identification, the amplification was performed with the primer set JDP1 (5′-GGCCCAGATCGTTTACCGTGAA-3′) and JDP2 (5′-TCTCACAAGCTGCTAGGGGAGTCA-3′) [43].

## 3. Results

A total of 97% (398/410) household stored water samples and 96% (392/410) biofilm swabs were collected in this study over the 5 month study period because a few household members were absent on the collection day and samples in a specific month could not be collected.

When viewed under an inverted microscope, 92% (366/398) of the water samples and 89.8% (352/392) of the biofilm samples contained FLA. The FLA observed were grouped into either trophozoites, round cysts, or *Acanthamoeba* spp. cysts based on the morphology only (Table 1). FLA were present in both the biofilm and stored container water as either a single form of FLA or in a combination (present as a cyst, trophozoites, or both in one sample).

Approximately 10% of presumptive positive samples were selected and sent for an 18S rRNA PCR and sequencing in Germany, of which the water samples comprised 11.5% (42/366) and the biofilm samples 9.7% (34/352) of the total number of samples. These samples included a total of 41 samples previously identified as trophozoites and 35 samples previously identified as presumptive *Acanthamoeba* cysts (Table 2).

Table 3 indicates the molecular identification of *Acanthamoeba* spp. and other FLA from the subset samples sent for the 18S rRNA PCR and sequencing. A total of 57.1% (16/28) of the water samples and 53.8% (7/13) of the biofilm samples tested with the FLA primer set yielded positive FLA results. With the specific *Acanthamoeba* spp. primer set, 42.9% (6/14) of the water samples and 19.0% (4/21) of the biofilm samples were FL-positive.

From the 76 samples sequenced, FLA (n = 50) were identified in only 33 of the samples. In addition, 37.0% (20/54) of *Vermamoeba* spp. and 29.6% (16/54) of *Acanthamoeba* spp. were the most abundant of the amoebae isolated (Table 4).

Table 5 lists the ARBs isolated from the stored water and confirmed with the 16S rRNA PCR and sequencing. Of the 398 water samples tested, 28.4% (113/398) yielded results for the presence of potential human bacteria with *Mycobacterium* spp. comprising the largest part of the isolates with 48.7% (55/113).

The 43 samples without positive FLA results either contained slime moulds or could not be amplified after repeated PCR attempts. The slime moulds belong to the group *Mycetozoa* that form part of the super-group *Amoebozoa* and comprise numerous amoebae [44] that can harbour bacteria [45].

## 4. Discussion

Although people are supplied with a safe water source, several practices can contaminate the water from the point of collection to the point of consumption (e.g., type of storage container used, cleanliness of the container, the handling of the water during collection and storage, water treatment practices at the household) [14,15]. However, most of these studies concentrated on the prevalence of indicator coliform bacteria and *Escherichia coli* whereas a few studies have investigated the survival of certain viruses and parasites in different storage water [46]. Very few studies have been conducted to provide information of the prevalence of FLA and potential human pathogenic bacteria in water stored at the household at the point of consumption.

In this study, all the households used wide-mouthed pre-used food containers to store their household water in (Figure 2). Several studies have proved that wide-mouthed domestic water storage vessels are more prone to contamination by unclean hands, dirty cups, and other water retrieval methods than screw-top containers (with their lids closed), which have a positive effect on the water quality [3,15,47,48,49,50]. It was also seen that the cup/jug (metal, glass, or plastic) used to retrieve the water from the storage vessel was often kept uncovered near the storage vessel and thus exposed to dust, flies, and unwashed hands, similar to the results of other studies in South Africa [3]. Other potential sources of groundwater contamination could be from agricultural activities and pit latrines [51] and a few studies have shown that the material of the containers could assist in the survival and transmission of potential pathogenic microorganisms [52,53].

Most of the FLA identified in this study were consistent with the environment from which the samples were taken (water and/or soil, or animal hosts). *Vermamoeba* spp. were one of the most abundant of the FLA isolated in this study (Table 4). *Vermamoeba vermiformis* has been isolated from various water sources both natural and man-made [54,55] as well as bat guano [56]. It has also proved to be a double threat as it may act as a potential pathogen in humans [57,58,59] and is also an important reservoir and vector for *Legionella pneumophila* [60], *Mycobacterium* spp. [58,61], and *Stenotrophomonas maltophilia* [54,62]. The other abundant FLA isolated in this study was *Acanthamoeba* spp. (Table 5); isolates containing genotype T4 have most frequently been isolated from human infections [63,64,65] because the T4 genotype is more transmissible and more virulent than other genotypes [66]. Other studies also reported *Acanthamoeba* spp. and *A. castellanii* str. *Neff* in soil samples [67]. The soil amoebae isolated from the samples included *Stenamoeba* spp. [68] and *Stenamoeba berchidia* [69]. Several FLA are known waterborne amoebae such as *Vexillefera westveldii* [70] and *Korotnevella hemistylolepis* [71]; *Flamella* spp. can be found in saltwater, freshwater, and soil samples [72]. Usually, FLA are ubiquitous and have been isolated from various natural sources such as freshwater, saltwater, dust, air, and soil [73,74]. Loret and Greub [75] collated data from recent studies and found that 62% of surface water and 71% of groundwater samples contained FLA. They often live in water–soil, water–air, and water–plant interfaces as well as on biofilms. Several FLA species attach to particulate matter to feed but others prefer to move in their planktonic phase [35,73,74,75]. Limitations of this study were the selection of the primer sets used for the identification of the FLA and the FLA enrichment cultivation method used that was selective in the isolation of the FLA from the water samples. Further studies with other isolation methods and different primers are needed to show the prevalence of other FLA in the water and biofilm samples.

Several bacteria are ingested by FLA and digested as a food source [35] whereas others have adapted to survive and grow within the amoebae by resisting the microbiocidal mechanisms of FLA and are termed amoeba-resistant bacteria [76]. These potential harmful bacteria are protected from adverse conditions until they excyst the amoebae [35,77]. These bacteria can persist within the FLA if the defence mechanisms of the amoebal cell are inefficient or impaired, which leads to a symbiotic relationship [78]. In a review by Thomas et al. [79], 102 of these bacterial species (all human pathogens and capable of survival in FLA) were identified. A few of the genera include *Achromobacter, Klebsiella, Mycobacterium, Legionella, Stenotrophomonas, Enterobacter*, and *Pseudomonas* [80] A wide range of potential human pathogenic bacteria were identified from the container-stored water samples in this study. Future studies could include determining accurately if these potential human bacteria species are ARBs retrieved from FLA. A few of the potential human pathogenic bacteria seen in this study are known human pathogens and are particularly fond of causing hospital-acquired infections such as *K. oxytoca* [80] and *S. maltophilia* [81] whereas several bacteria identified in this study are found naturally in the environment such as *E. amnigenus* [82] and *M. salmoniphilum* [83] in animals and *P. kilonensis* [84] and *P. koreensis* [85] in soil. Numerous *Mycobacterium* spp. were isolated from the stored water samples. Mycobacterial species can live and grow in amoebae for years in a truly symbiotic relationship [86]; it was also proven that *Acanthamoeba castellanii* can amass several different mycobacterial species within their cytoplasm [87].

## 5. Conclusions

The results from this study showed the prevalence and co-existence of FLA and potential harmful bacteria which could be a health concern, that were found in household-stored water and containers, highlighting the need for improved hygiene education and intervention programmes on water collection and treatment practices at a household level. Many of the FLA and potential harmful bacterial spp isolated from the stored drinking water and the container biofilms were potential pathogens and a health risk to vulnerable individuals who could drink the water without any prior treatment. In addition, to our knowledge, this is the first study in South Africa to isolate FLA and potential harmful bacteria present in storage container biofilms as well as identify them at a species level. Several recommendations can, therefore, be made from the results of this study. Firstly, further studies are needed to examine the prevalence, behaviour, and possible disinfection of FLA in both the planktonic and biofilm states. Secondly, further studies are needed to examine the removal of these organisms from the storage containers to protect the communities that rely on the water, especially with the increase in demand for groundwater in semi-arid areas such as South Africa. Thirdly, further studies are needed to examine the causal links between the organisms found and the clinical symptoms presented by the users of the water (if any) as well as exploring the possibility of changing conventional water testing by means of indicator organisms alone to include more organisms that have proven to be emerging pathogens. Lastly, more studies are needed to assess the effect of global warming on the survival and distribution of pathogenic amoebae and to determine the microbiomes inside these amoebae to increase our understanding of the health risks of amoeba protection strategies in drinking water.

## Figures and Tables

**Figure 1 biology-10-01228-f001:**
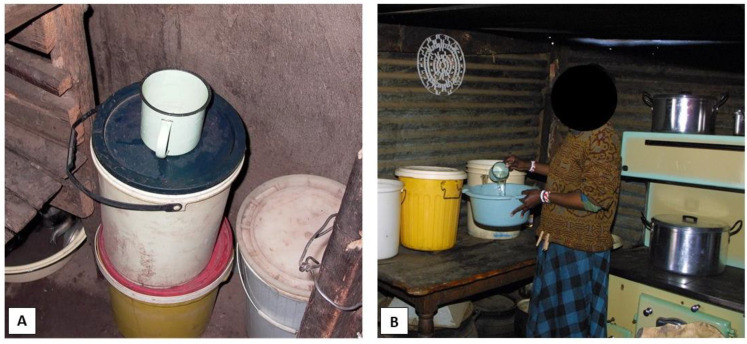
Water storage container in study group. (**A**) Typical water storage buckets in the households with a retrieval cup on top. (**B**) A female study participant decanting water from the water storage container.

**Figure 2 biology-10-01228-f002:**
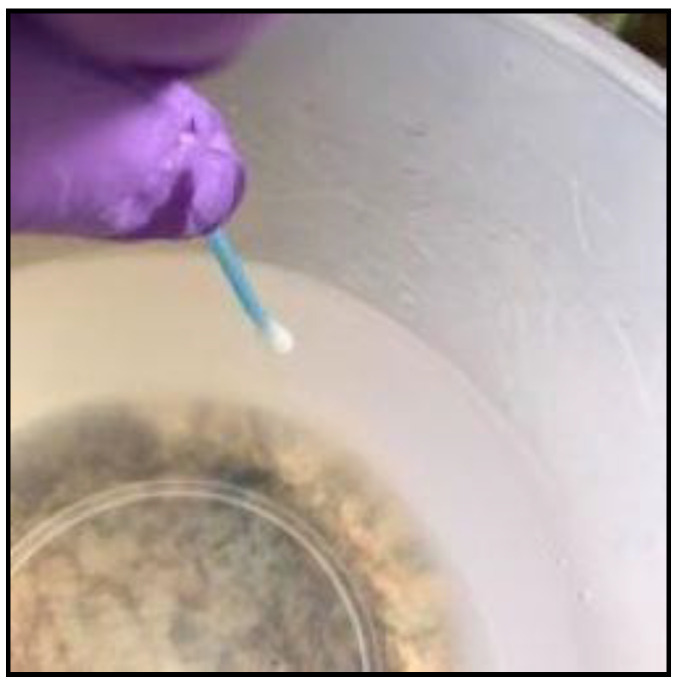
A swab taken from water storage container.

**Table 1 biology-10-01228-t001:** Presumptive FLA identification by inverted microscope.

Presumptive Identification of FLA Types	Water in Storage Container (n = 398)	Biofilm in Storage Container (n = 392)
Trophozoites	*Acanthamoeba* cysts	Round cysts
26.9% (107/398)	89.2% (355/398)	32.2% (128/398)
13.8% (54/392)	79.6% (312/392)	48.5% (190/392)

**Table 2 biology-10-01228-t002:** FLA selected for an 18S rRNA PCR and sequencing.

FLA Type	Water (n = 42)	Biofilm (n = 34)
Trophozoites (n = 42)	28	13
Presumptive *Acanthamoeba* (n = 43)	14	21

**Table 3 biology-10-01228-t003:** Samples positive for FLA according to the sample types and primer sets.

Primer Set Used	Water in Storage Container (n = 42)	Biofilm in Storage Container (n = 34)
Ami6F1, Ami6F2, Ami9R (FLA) [42]	38.1% (16/42)	20.6% (7/34)
JDP1, JDP2 (*Acanthamoeba* spp.) [44]	14.3% (6/42)	11.8% (4/34)

**Table 4 biology-10-01228-t004:** FLA identified using 18S rRNA sequencing from storage container biofilm and water samples.

FLA Types Isolated	Total nr of FLA Isolated (n)	FLA Isolated form Stored Water	FLA Isolated from Biofilm
*Amoebozoa* spp.	5	3	2
*Acanthamoeba* spp.	5	3	2
Genotype T3	1	1	0
Genotype T4	5	3	1
Genotype T15	1	1	0
Genotype T16	4	2	2
*Flamella* spp.	2	2	0
*F. fluviatalis*	1	1	0
*Korotnevella* spp.			
*Korotnevella hemistylolepis*	1	0	1
*Lobosea* spp.	1	1	0
*Stenamoeba* spp.	2	2	0
*S. berchidia*	1	0	1
*Vermamoeba* spp.			
*V. vermiformis*	20	16	4
*Vexillifera* spp.			
*V. westveldii*	1	1	0
Total	50	36	14

**Table 5 biology-10-01228-t005:** Potential human bacteria pathogens isolated in stored household water samples.

Potential Human Pathogen Bacteria Isolated		
*Pseudomonas* spp.: *Pseudomonas fluorescens* *Pseudomonas geniculata* *Pseudomonas kilonensis* *Pseudomonas koreensis* *Pseudomonas poae* *Pseudomonas tremae* *Pseudomonas vancouverensis* *Pseudomonas poae*/*tolaasii* *Pseudomonas putida*	*Arthobacter* spp.: *Arthrobacter nicotinovorans*	*Mycobacterium* spp.: *Mycobacterium chlorophenolicum* *Mycobacterium chubuense* *Mycobacterium elephantis* *Mycobacterium fallax* *Mycobacterium farcinogenes* *Mycobacterium florentinum* *Mycobacterium gilvum* *Mycobacterium intermedium* *Mycobacterium llatzerense* *Mycobacterium noviomagense* *Mycobacterium pallens* *Mycobacterium poriferae* *Mycobacterium psychrotolerans* *Mycobacterium rhodesiae* *Mycobacterium salmoniphilum* *Mycobacterium smegmatis* *Mycobacterium tokaiense* *Mycobacterium triplex*
	*Caulobacter* spp.: *Caulobacter segnis*	
	*Klebsiella* spp.: *Klebsiella oxytoca* *Klebsiella variicola*	
*Enterobacter* spp.: *Enterobacter amnigenus* *Enterobacter asburiae* *Enterobacter cancerogenus* *Enterobacter kobei* *Enterobacter ludwigii*	*Paenibacillus* spp.: *Paenibacillus validus*	
	*Pragia* spp.: *Pragia fontium*	*Microbacterium* spp.: *Microbacterium* spp. *Microbacterium oxydans* *Microbacterium paraoxydans*
*Achromobacter* spp.: *Achromobacter insolitus* *Achromobacter marplatensis* *Achromobacter spanius*	*Rhodococcus* spp.: *Rhodococcus erythropolis*	
	*Sarratia* spp.: *Serratia ureilytica*	
	*Stenotrophomonas* spp.: *Stenotrophomonas maltophilia*	
Mixed bacterial spp.: *Alcaligenes faecalis* and *Achromobacter marplatensis*		

## Data Availability

The data presented in this study are available on request from the corresponding author.

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
