# Peer review of "Co-Existence of Free-Living Amoebae and Potential Human Pathogenic Bacteria Isolated from Rural Household Water Storage Containers"

_biology, 2021, doi:10.3390/biology10121228_

Round 1

Reviewer 1 Report

This is an interesting contribution, dedicated to important topic and valuable both form the fundamental and practical point of view. However I have a number of question/comments to the general strategy of the study as well as to the obtained results and the ms itself.

Methods need to be better detailed. Brief description of procedures used should be provided. In the present version author refer to other papers for methods, but these papers (in turn) do not contain detailed description but refers to further papers, and this makes understanding difficult. If one follow this chain to the end, he discovers rather trivial (and selective) method for amoebae isolation. The isolation method on NNA plates is much older than any of papers cited in the ms (dated back to 1922).All these methods were described in details by F. Page in 1976 and 1988, if not to dig that deep. 

The enrichment cultivation is  a selective method. It recovers only a fraction of amoebae diversity. Further, PCR amplification is also selective. "Ami" primers are far not that universal as it is stated, so it is the further step of selection. Authors should clearly indicate this. In fact, the set of recovered species votes for the same - mostly these are Acathamoeba and Vermamoeba strains, other species were recovered in few cases. So the result is a fraction of FLA species potentially inhabiting biofilms. I am sure (from my own experience) that in such biofilms there are many more amoebae species.

I cannot understand how species like Entamoeba invadens could form "a colony" on NNA plate (as it follows from the description of methods). This is an anaerobic, amitochondriate organism. Probably, cysts occurred in the DNA samples. Or cysts were filtered out from the water samples - this is not clear from the text. These cysts could occur in the water a s a results of contamination (with faeces?). May I suggest not to list them as species "found in the biofilm". Line 212 - Entamoeba is not a FLA!

Table 6. It is not clear why ARB are termed associated with amoebae. Co-occurred? This is a big question, if bacteria are really associated with amoeba cells or just live in the same environment.

The "Discussion" section looks like an independent review, not directly related to the results, but telling some general things of FLA and ARB (the second Intro). This makes reading difficult. 

It is clear that there is not much sense to provide images of all strains found. But it would be useful to provide images of rare species (like Stenamoeba or Korotnevella) and images of non-identified isolates. This could be very helpful in future.

Author Response

Reviewer 1:

This is an interesting contribution, dedicated to important topic and valuable both form the fundamental and practical point of view. However, I have a number of question/comments to the general strategy of the study as well as to the obtained results and the ms itself.

The authors are thankful to the reviewer for the comments/questions – it assists us in improving the manuscript

Methods need to be better detailed. Brief description of procedures used should be provided. In the present version author refer to other papers for methods, but these papers (in turn) do not contain detailed description but refers to further papers, and this makes understanding difficult. If one follow this chain to the end, he discovers rather trivial (and selective) method for amoebae isolation. The isolation method on NNA plates is much older than any of papers cited in the ms (dated back to 1922). All these methods were described in details by F. Page in 1976 and 1988, if not to dig that deep.

The methods were better described and more detailed as requested with more references

The enrichment cultivation is a selective method. It recovers only a fraction of amoebae diversity. Further, PCR amplification is also selective. "Ami" primers are far not that universal as it is stated, so it is the further step of selection. Authors should clearly indicate this. In fact, the set of recovered species votes for the same - mostly these are Acathamoeba and Vermamoeba strains, other species were recovered in few cases. So the result is a fraction of FLA species potentially inhabiting biofilms. I am sure (from my own experience) that in such biofilms there are many more amoebae species.

The authors agree with the reviewer that the enrichment method was a selective method.  As there are several methods to use for isolation of amoebae and this add to costs for a project, we had to take note of this and our collaborators in Germany preferred the enrichment process.  They also selected the primers we used for identification.  We also removed the word “universal” because it creates confusion.  We meant it is a generally used primer set seen in many publications.  We did make the statement in the discussion that the primer sets used for the identification of the FLA and the enrichments cultivation was selective methods and more studies with other isolation methods and different primers is needed to assess the presence of FLA in water and biofilm samples.

I cannot understand how species like Entamoeba invadens could form "a colony" on NNA plate (as it follows from the description of methods). This is an anaerobic, amitochondriate organism. Probably, cysts occurred in the DNA samples. Or cysts were filtered out from the water samples - this is not clear from the text. These cysts could occur in the water a s a results of contamination (with faeces?). May I suggest not to list them as species "found in the biofilm". Line 212 - Entamoeba is not a FLA!

The authors took note of this and removed all the Entamoeba results and discussion points in the manuscript

Table 6. It is not clear why ARB are termed associated with amoebae. Co-occurred? This is a big question, if bacteria are really associated with amoeba cells or just live in the same environment.

The authors decided that it is better to remove Table 6 because it creates confusion and it does not make any impact in the context of the manuscript’s main theme

The "Discussion" section looks like an independent review, not directly related to the results, but telling some general things of FLA and ARB (the second Intro). This makes reading difficult.

The authors removed some of the content in the discussion and we hope it reads better that before

It is clear that there is not much sense to provide images of all strains found. But it would be useful to provide images of rare species (like Stenamoeba or Korotnevella) and images of non-identified isolates. This could be very helpful in future.

We agree but do not have clear images due to the camera that was giving problems

Reviewer 2 Report

This study looked at the presence of free-living amoeba and potentially pathogenic bacteria from domestic water used by dwellers of “informal settlements” in South Africa. The manuscript is well written and clear. The major problem is the use of the term “amoeba-resistant bacteria”.  The bacteria in question were just isolated from the same water sample. There is no connection between the amoebae and the bacteria other than that, although of course it is likely that some of the bacteria interact with some of the amoeba in these samples as food organisms or predators of amoeba.  The use of the term should be removed unless the possibility of amoebal resistance is being discussed.

Water samples were taken between March and July. Was the temperature of the borehole water measured throughout this period? Or the temperature of the water from the samples? Naegleria fowleri was not detected in these samples which is important since this is a dangerous human pathogen that requires temperatures to be consistently about 30oC to allow it to become abundant.

No information as to the geographic area of the study. If there are sensitivity about the exact location, perhaps some vague description such as “30 km from Pretoria” could be used. Some indication of the area is needed.

The primers used Ami6F1, Ami6F2 and Ami9R usually amplify Amoebozoans but seem less efficient for Naegleria (Delafont et al, 2013; Muchesa et al, 2016). The authors should discuss the possible role of primer bias in the range of FLA discovered by their 18S PCR.

The amoeba-resistant bacteria term should be used more cautiously as a bacterium that resist one amoeba may not resist another. Also, there was no test carried out to determine if the particular bacterial strains isolated here and labelled as ARBs here were resistant to any amoebae?  At least no such tests were mentioned. What is the basis for the label ARB on line 102 and table 5 for example?  Perhaps it would be better to describe these bacteria as potential ARBs? In most places within the text the term ARB should be changed to potential human pathogens since this is the more relevant issue.

The title of the paper too should be changed since there was no test to ascertain if the bacteria isolated were resistant to the amoeba. Perhaps just remove the words “amoeba-resistant”?

Table 4 is a bit confusing as Acanthamoeba castellanii str. Neff is a T4?

Line 25. not all Acanthamoeba T4 strains are pathogenic to humans. Better to re-write as “including Acanthamoeba genotype T4 strains some of which are known to be pathogenic to humans”

Line 36. It would be helpful to mention what “StatsSA” is?

Line 108 How much of the 1litre sample was filtered? All of it?

Line 185 Escherichia coli should be in italics

Line 241 “genera” rather than “genuses”

Delafont, V., Brouke, A., Bouchon, D., Moulin, L. and Héchard, Y., 2013. Microbiome of free-living amoebae isolated from drinking water. Water research47(19), pp.6958-6965.

Muchesa, P., Leifels, M., Jurzik, L., Barnard, T.G. and Bartie, C., 2016. Free-living amoebae isolated from a hospital water system in South Africa: a potential source of nosocomial and occupational infection. Water Science and Technology: Water Supply16(1), pp.70-78.

Author Response

Reviewer 2:

This study looked at the presence of free-living amoeba and potentially pathogenic bacteria from domestic water used by dwellers of “informal settlements” in South Africa. The manuscript is well written and clear. The major problem is the use of the term “amoeba-resistant bacteria”. The bacteria in question were just isolated from the same water sample. There is no connection between the amoebae and the bacteria other than that, although of course it is likely that some of the bacteria interact with some of the amoeba in these samples as food organisms or predators of amoeba.  The use of the term should be removed unless the possibility of amoebal resistance is being discussed.

The term “amoeba resistant bacteria” was changed in the manuscript to “potential human pathogenic bacteria” and it was used in the context of co-existence with FLA

Water samples were taken between March and July. Was the temperature of the borehole water measured throughout this period? Or the temperature of the water from the samples? Naegleria fowleri was not detected in these samples which is important since this is a dangerous human pathogen that requires temperatures to be consistently about 30oC to allow it to become abundant.

Water sample temp was taken each time a water sample was collected.  In the 5 months, the average of water samples was 20oC in the storage containers and 21oC in the sources.    

No information as to the geographic area of the study. If there are sensitivity about the exact location, perhaps some vague description such as “30 km from Pretoria” could be used. Some indication of the area is needed.

The study area was included as close as possible without directly naming the area – it is a sensitive topic and therefore specifics cannot be made public.

The primers used Ami6F1, Ami6F2 and Ami9R usually amplify Amoebozoans but seem less efficient for Naegleria (Delafont et al, 2013; Muchesa et al, 2016). The authors should discuss the possible role of primer bias in the range of FLA discovered by their 18S PCR.

  • Delafont, V., Brouke, A., Bouchon, D., Moulin, L. and Héchard, Y., 2013. Microbiome of free-living amoebae isolated from drinking water. Water research, 47(19), pp.6958-6965.
  • Muchesa, P., Leifels, M., Jurzik, L., Barnard, T.G. and Bartie, C., 2016. Free-living amoebae isolated from a hospital water system in South Africa: a potential source of nosocomial and occupational infection. Water Science and Technology: Water Supply, 16(1), pp.70-78.

We did add a sentence in the discussion on the bias on isolation methods and primer sets used.

The amoeba-resistant bacteria term should be used more cautiously as a bacterium that resist one amoeba may not resist another. Also, there was no test carried out to determine if the particular bacterial strains isolated here and labelled as ARBs here were resistant to any amoebae? At least no such tests were mentioned. What is the basis for the label ARB on line 102 and table 5 for example?  Perhaps it would be better to describe these bacteria as potential ARBs? In most places within the text the term ARB should be changed to potential human pathogens since this is the more relevant issue.

The authors took note of this and took out the term ““amoeba resistant bacteria” and changed it in the manuscript to “potential human pathogenic bacteria”.

The title of the paper too should be changed since there was no test to ascertain if the bacteria isolated were resistant to the amoeba. Perhaps just remove the words “amoeba-resistant”?

The title was changed as reflected in the manuscript

Table 4 is a bit confusing as Acanthamoeba castellanii str. Neff is a T4?

The table content was corrected and the T4 counts added correctly

Line 25. not all Acanthamoeba T4 strains are pathogenic to humans. Better to re-write as “including Acanthamoeba genotype T4 strains some of which are known to be pathogenic to humans”

This was done as per request of reviewer

Line 36. It would be helpful to mention what “StatsSA” is?

This was done as per request of reviewer

Line 108 How much of the 1litre sample was filtered? All of it?

This was added in the materials and methods section as per request of reviewer

Line 185 Escherichia coli should be in italics

This was done as per request of reviewer

Line 241 “genera” rather than “genuses”

This was done as per request of reviewer

Reviewer 3 Report

General Comments

  1. The manuscript reports a survey of stored household water for amoeba and the bacteria associated with amoebae (amoeba-resistant bacteria). As climate change has been accompanied with reduced and erratic water supplies, stored-water is becoming essential in some parts of the world.
  2. The information is of value for it identifies a possible important source of waterborne amoeba and bacterial pathogens. Knowledge that home-stored water is a likely habitat of pathogenic microorganisms, here namely both bacteria and amoeba, can be used by governments to provide guidance and by homeowners to prepare, clean and disinfect containers for home-stored water.
  3. The Materials and Methods can be employed to guide the cleaning and disinfection of home water storage containers to reduce the possibility of contamination and infection.
  4. The manuscript is well-written, the methods clear, and the data of significant value.

Author Response

Reviewer 3:

The manuscript reports a survey of stored household water for amoeba and the bacteria associated with amoebae (amoeba-resistant bacteria). As climate change has been accompanied with reduced and erratic water supplies, stored-water is becoming essential in some parts of the world.

Acknowledged by authors

The information is of value for it identifies a possible important source of waterborne amoeba and bacterial pathogens. Knowledge that home-stored water is a likely habitat of pathogenic microorganisms, here namely both bacteria and amoeba, can be used by governments to provide guidance and by homeowners to prepare, clean and disinfect containers for home-stored water.

Acknowledged by authors

The Materials and Methods can be employed to guide the cleaning and disinfection of home water storage containers to reduce the possibility of contamination and infection.

Acknowledged by authors

The manuscript is well-written, the methods clear, and the data of significant value.

The authors are thankful to the reviewer for the positive comments